# Small Angle X-ray Scattering Sensing Membrane Composition: The Role of Sphingolipids in Membrane-Amyloid β-Peptide Interaction

**DOI:** 10.3390/biology11010026

**Published:** 2021-12-25

**Authors:** Rita Carrotta, Maria Rosalia Mangione, Fabio Librizzi, Oscar Moran

**Affiliations:** 1Consiglio Nazionale delle Ricerche, Istituto di Biofisica, Via Ugo La Malfa 153, 90146 Palermo, Italy; mariarosalia.mangione@ibf.cnr.it (M.R.M.); fabio.librizzi@ibf.cnr.it (F.L.); 2Consiglio Nazionale delle Ricerche, Istituto di Biofisica, Via De Marini 6, 16149 Genova, Italy; oscar.moran@cnr.it

**Keywords:** Aβ, GM1, sphingomyelin, SAXS, large unilamellar vesicles, rafts

## Abstract

**Simple Summary:**

The early impairments in Alzheimer’s disease are related to neuronal membrane damage. Different lipids are present in biological membranes, playing relevant physiological roles. Some of them, such as sphingomyelin, cholesterol, and ganglioside GM1, interact with each other and, importantly, with the Aβ peptide. Here, these interactions are studied using small angle X-ray scattering in model membrane systems, such as large unilamellar liposomes. This technique gives information on the width of the bilayer and reveals structural differences due to the different lipid compositions, as well as some small differences due to the presence of the Aβ peptide. The analysis highlights the concentration-dependent effect of GM1 on membrane thickness and the interaction with the Aβ-peptide, together with the inhibiting effect that the presence of sphingomyelin has on the GM1–Aβ interaction.

**Abstract:**

The early impairments appearing in Alzheimer’s disease are related to neuronal membrane damage. Both aberrant Aβ species and specific membrane components play a role in promoting aggregation, deposition, and signaling dysfunction. Ganglioside GM1, present with cholesterol and sphingomyelin in lipid rafts, preferentially interacts with the Aβ peptide. GM1 at physiological conditions clusters in the membrane, the assembly also involves phospholipids, sphingomyelin, and cholesterol. The structure of large unilamellar vesicles (LUV), made of a basic POPC:POPS matrix in a proportion of 9:1, and containing different amounts of GM1 (1%, 3%, and 4% mol/mol) in the presence of 5% mol/mol sphingomyelin and 15% mol/mol cholesterol, was studied using small angle X-ray scattering (SAXS). The effect of the membrane composition on the LUVs–Aβ-peptide interaction, both for Aβ_1–40_ and Aβ_1–42_ variants, was, thus, monitored. The presence of GM1 leads to a significant shift of the main peak, towards lower scattering angles, up to 6% of the initial value with SM and 8% without, accompanied by an opposite shift of the first minimum, up to 21% and 24% of the initial value, respectively. The analysis of the SAXS spectra, using a multi-Gaussian model for the electronic density profile, indicated differences in the bilayer of the various compositions. An increase in the membrane thickness, by 16% and 12% when 2% and 3% mol/mol GM1 was present, without and with SM, respectively, was obtained. Furthermore, in these cases, in the presence of Aβ_40_, a very small decrease of the bilayer thickness, less than 4% and 1%, respectively, was derived, suggesting the inhibiting effect that the presence of sphingomyelin has on the GM1–Aβ interaction.

## 1. Introduction

The etiology of Alzheimer’s disease (AD) remains an unsolved issue. The main hallmark of AD, besides the evident neurodegenerative symptoms, is the formation of extracellular and intracellular protein aggregates, composed of amyloid β-peptide (Aβ) and tau protein, respectively. However, the correlation between the amount of protein deposits and the progression of the disease is controversial, although the most popular theory to explain the disease remains the amyloid hypothesis [1,2]. This recognizes the onset of the degeneration in the Aβ fibril deposits and the intracellular aggregation of the tau protein as a secondary effect. The involvement of protein misfolding in AD supports the hypothesis that small toxic Aβ species, formed in vivo and escaping the machinery of the protein quality control in the cell, represent the starting point of neuronal disfunction [3,4]. This hypothesis can explain the difficulty in diagnosing the early stages of the disease. In fact, due to their nature, the detection of these oligomeric species, prone to inducing the formation of fibril aggregates, is a challenging task; they are nanometer sized and metastable. These oligomers are very reactive species in the cellular environment, and able to interact with membrane components, such as proteins and lipids.

Aβ-peptides are derived from the processing of the trans-membrane amyloid precursor protein (APP). Its processing by secretase enzymes produces peptides of different lengths, released in the extracellular medium, and having different propensities to aggregate. Along with deposition of plaques, observed in the neuronal extracellular space, there is also evidence of intracellular accumulation of Aβ, which may be responsible for the early synaptic dysfunction and memory impairment [5,6]. The presence of intracellular Aβ is supported by the fact that APP can be found in the neuronal plasma membrane but also in the other membranes of the intracellular compartments, and that secretases are present in different subcellular environments. Furthermore, some studies have shown that Aβ can be internalized from the extracellular space [6]. From these considerations, the interaction of Aβ with cell membranes is considered one of the possible causes of early neurologic disfunction. In fact, toxic Aβ species may cause some specific perturbations to the lipid matrix, from a physicochemical and structural point of view, disturbing the correct function of some intrinsic membrane proteins, such as receptors and channels; on the other hand, some lipid unbalances can cause the formation of toxic Aβ species close to the membrane, with consequences on the functional properties of the bilayer and on protein aggregation. In some studies, a preferential interaction of Aβ species with cholesterol and sphingolipids was reported [7,8,9,10]. These interactions occur in membrane regions, named ‘lipid rafts’, with a high content of sterols and sphingolipids. Lipid rafts are essential for cell reactions and signaling pathways [11]. A preferential role of GM1 in the Aβ–membrane interaction has been reported [12]. Studies at different GM1:Aβ ratios produced different outcomes, inhibitory or acceleratory, with aggregation of Aβ in the presence of a model membrane system containing GM1 [8,10].

In this work we investigated the interaction of Aβ peptide with the sphingolipids GM1 and sphingomyelin (SM) in a lipid bilayer model, containing phospholipids (POPC and POPS) and cholesterol. We used large unilamellar vesicles (LUV) in order to study: (1) whether and how the presence of amyloid peptides affects the bilayer structure, and (2) how the composition of the membrane affects the potential interaction, as resolved with small angle X-ray scattering (SAXS). SAXS is a good method for highlighting structural features on the length-scale of nanometers, such as in proteins, lipid bilayers, or their complexes [13,14,15,16]. The SAXS spectrum results from the spatially different electronic density distribution with respect to the solvent; therefore, it contains information about the shape and structure of the scattering particles. The dependence on GM1 concentration was studied in the presence of a constant concentration of sphingomyelin and cholesterol. The concentration range of GM1 was chosen to mimic physiological conditions in the brain membranes (GM1 less than 5% mol/mol). Moreover, the Aβ concentration for the interaction studies was adjusted to 50 μM for Aβ_1–40_ and to 10 μM for Aβ_1–42_. These concentrations of amyloid peptides did not lead to uncontrolled aggregation of Aβ and gave a longer aggregation lag phase compared to the time of the SAXS experiment. In a time scale where no aggregation was observed, the comparison of results for bilayer matrices, with and without SM, showed competition of the two sphingolipids, GM1 and SM, in the interaction of both variants, Aβ_1–40_ and Aβ_1–42_, with the membrane. 

## 2. Experimental Methods

### 2.1. Sample Preparation

LUV were prepared by extrusion, and their interaction with the Aβ-peptide was studied.

#### 2.1.1. Materials

Monounsaturated phospholipids (16:0, 18:1) POPC (cat. P3017) and POPS (cat. 51581), monosialoganglioside from bovine brain GM1 (cat. G7641), chicken egg yolk sphingomyelin, ESM (cat. S0756), and sheep’s wool cholesterol (cat. C8667) were purchased from Sigma-Aldrich (Milan, Italy) and used without any further purification. Amyloid β-peptide 1–40 (Aβ_1–40_) was purchased from AnaSpec (Freemont, CA, USA) and, prior to use, was pretreated and re-lyophilized as described elsewhere [17], and characterized by thioflavin T kinetic tests, AFM, and light scattering to verify their uniformity with previous studies [7,18,19,20]. Interaction studies were performed with a final total lipid vesicle and Aβ_1–40_ concentration of 15 mM and 50 μM, respectively, to reach a molar ratio of 300:1, as reported in a previous study [13]; for Aβ_1–42_ the molar ratio lipid:peptide was 1500:1.

#### 2.1.2. LUV Preparation

Different LUVs were made from films, obtained by thin layer evaporation of chloform/methanol (9:1) lipids solutions, prepared by suitably combining the different lipid components, according to Table 1. After complete film hydration with 50 mM phosphate buffer with 20 mM NaCl (pH 7.4), obtained using freeze–thawing cycles, the liposome solutions were extruded through a polycarbonate membrane filter with 100 nm pore (Avestin, Mannheim, Germany), with 31 extrusion steps. Samples were prepared considering a lipid mass at a final concentration of 15 mg/mL. To assess whether mass loss had occurred during the extrusion step, the final concentration of the LUV was checked using static light scattering (SLS), as described elsewhere [21]. Liposomes samples, if necessary, were concentrated through Centricon 100 KDa centrifuge filters before the addition of the Aβ peptides. Dynamic light scattering (DLS) was used to verify the hydrodynamic size distribution of the LUV and, therefore, the extrusion quality. Scattered light intensity and the time autocorrelation function were measured using a Brookhaven BI-9000 correlator and a 100 mW solid-state laser at *λ* = 532 nm. Measurements were performed at *Q* = 22.3 μm^−1^. The field autocorrelation functions obtained by DLS were analyzed using a smoothing constrained regularization method [22]. The GM1-free matrices with different concentrations of sphingomyelin were named B1 (PC:PS:Chol), B2 (PC:PS:Chol:SM5%), and B3 (PC:PS:Chol:SM10%).

#### 2.1.3. Aβ Preparation

The synthetic peptide Aβ (Anaspec) was solubilized in 5 mM NaOH (Sigma-Aldrich), pH 10, and lyophilized, according to Fezoui and collaborators [17]. The lyophilized peptide was then dissolved in a 50 mM sodium phosphate buffer with 20 mM NaCl, and sequentially filtered through Millex LG 0.22 μm and Amicon 20 nm filters (Millipore, Milan, Italy). The concentration was determined by absorbance at 276 nm (ε_276 nm_ = 1390 M^−1^cm^−1^). Aβ_1–40_ and Aβ_1–42_ were prepared at approximately 70 μM and 20 μM, respectively. For the interaction studies, freshly dissolved stock solutions of Aβ_1–40_ and Aβ_1–42_ were mixed with the vesicles, to reach the final concentrations of 50 μM and 10 μM, respectively, in both cases with 15 mM of LUVs.

#### 2.1.4. SAXS Method and Analysis 

The small-angle X-ray scattering (SAXS) spectra of LUV were collected at the BL11 beam line of the ALBA Synchrotron Light Facility (Barcelona, Spain). The scattered radiation was recorded using a two-dimensional CCD detector. The sample–detector distance of 2.39 m covered the momentum transfer interval 0.14 < *s* < 4.5 nm^−1^ (*s* = 4π sin (θ)/λ, where 2θ is the scattering angle and λ = 0.128 nm is the X-ray wavelength), the optical path of the X-ray through the sample was approximately 3 mm. Data were collected from the sample kept at 20 °C. For each sample, we recorded 40 spectra of 5 s each, corresponding to a total of 3.3 min of data acquisition. The CCD camera images taken from the random orientation of vesicles were integrated radially, resulting in a so-called ‘*I-s* plot’: a one-dimensional profile of X-ray intensity *I*(*s*) versus scattering vector *s*. The comparison of the 40 successive exposures of the acquisition experiment indicated no changes in the scattering patterns, i.e., no measurable radiation damage to the lipid samples. When the peptide was present, the absence of changes in the 40 exposures excluded the presence of aggregation phenomena. The buffer scattering data, tested before and after each corresponding measured sample, were averaged and used to subtract the background.

#### 2.1.5. SAXS Data Analysis 

By analyzing the SAXS spectra, it was possible to deduce the structure of the membranes by obtaining the electronic density profiles. In these experiments, the measured X-ray intensity was averaged over a polydisperse vesicle population and according to Debye scattering theory [23]; by considering the high dilution limit and the wave vector range studied (*s* > 0.1 nm^−1^), a simplified relation can be used as a fitting expression to the data:(1)$$\langle I\left(s\right)\rangle \propto \langle F{\left(s\right)}^{2}\rangle$$
where *F*(*s*) is the bilayer form factor and the proportionality is such that the number of scattering particles is included in the arbitrary instrumental scaling. The form factor, *F*(*s*), is the Fourier transform of the electronic density *ρ*(*r*) of the bilayer. The electronic density model of the membrane is used to fit its Fourier transform to the experimental SAXS spectra.

The electronic density of the vesicle wall can be described by the overlap of the number of concentric Gaussian shells, as described in previous works [24,25,26,27,28]. Figure 1a represents the seven Gaussians modelling the electronic density profile. Two Gaussians with positive amplitude represent the headgroups of the two leaflets (in and out), while a Gaussian with negative amplitude, symmetrically positioned in the center of the bilayer, accounts for the phospholipid tail region (tail); two Gaussians, indexed as inner and outer, are considered decorations, due to asymmetric groups (such as the GM1 head and/or interacting proteins in this case); the last two Gaussians (ch_in and ch_out) correct for the asymmetric cholesterol positioning. According to this model, the electronic density of the bilayer profile, as a function of the distance *r,* is given by:(2)$$\rho \left(r\right)=\sum_{k}\rho_{k}\exp\left[\frac{-{\left(r-\delta_{k}\right)}^{2}}{2\sigma_{k}^{2}}\right]$$
where *δ**_k_* is the peak position, *ρ**_k_* is the amplitude, and *σ**_k_* the width, with *k* indicating in, *out*, tail, inner, outer, ch_in, and ch_out.

Thus, as shown in Figure 1b, we define $\varepsilon_{k}=\delta_{k}-R$, and $\varepsilon_{k}=0$ for *k* corresponding to the tail gaussian. ch_in and ch_out are forced to stay into the in-tail and tail-out space, respectively, and the positions of the inner and outer gaussians are also forced by $\varepsilon_{\mathrm{inner}}\leq \varepsilon_{\mathrm{in}}$ and $\varepsilon_{\mathrm{outer}}\geq \varepsilon_{\mathrm{out}}$ in a manner that they can, to some extent, penetrate the phospholipid bilayer space. The membrane thickness, *d*, can be estimated from the electron distribution as the region into the two shells, inner and outer, fixing a threshold $\rho_{el}=0.05$.

For a perfectly spherical, radially symmetric vesicle composed of *n* Gaussian shells, a normalized ensemble average over *F*(*s*)^2^ can be used as a form factor expression [24,25,26]:(3)$$\langle F{\left(s\right)}^{2}\rangle =\zeta \left\{\frac{1}{s^{2}}\underset{k,k\prime }{\sum }\left(R+\epsilon_{k}\right)\left(R+\epsilon_{k^{\prime }}\right)\rho_{k}\rho_{k^{\prime }}\sigma_{k}\sigma_{k^{\prime }}\times \exp\left[-s^{2}\left(\frac{\sigma_{k}^{2}+\sigma_{k\prime }^{2}}{2}\right)]\cos[s\left(\epsilon_{k}-\epsilon_{k^{\prime }}\right)\right]\right\}$$
where ζ is a proportionality factor. This equation describes well the SAXS intensity in the region 0.1 nm^−1^ < *s* < 10 nm^−1^, where the intra-bilayer features dominate the scattering curve [24]. The non-linear, least-squares Levenberg–Marquardt fitting (NLSF) algorithm was used to obtain the best data fitting function (IgorPro, Wavemetrics, Lake Oswego, OR, USA).

The electronic density profile of the lipid vesicle wall was obtained with high accuracy fitting of the model over the experimental data.

This method also allows having a good estimate of the average vesicle radius.

## 3. Results

### 3.1. Characterization of the Liposomes

The liposomes were characterized by dynamic light scattering (DLS). The characterization was performed in three independent preparations, and no significant differences in the hydrodynamic sizes for the different matrices could be appreciated (Appendix A). On the contrary, SAXS spectra showed evident differences, due to the different compositions of the lipids. In Figure 2, the SAXS spectra of the different matrices are compared to highlight the role of the sphingolipids in the liposomes, both ganglioside, GM1, and sphingomyelin, SM. Figure 2a reports the comparison of the matrices with 5% mol/mol SM at different concentrations of GM1, from 0 to 4%.

The presence of GM1 produced a shift of the main maximum peak towards lower *s*-values, together with an intensity decrease, with those effects accompanied by a shift of the first node towards higher *s*-values and a decrease of its strength, as can be seen in Figure 2a,c, which reports the spectra relative to B1 and B2, with or without GM1. Taken together these features evidence a decrease of contrast (lower intensity) and an increase in the fluctuations of the bilayer (broadening of the spectrum) and in its thickness (lower value for s_Max_).

Figure 2b shows the effect of SM on the matrix. In this case, the bilayer is stiffer the more sphingomyelin is present, as demonstrated by the deepening of the first node and, at the same time, by the decrease in the contrast. In the corresponding lower panels, reporting the electron density profile, it can be seen that the presence of GM1 correlates in the outside leaflet of the vesicle with the need to add a Gaussian decoration in the analysis, being interpreted as describing the GM1 polar head, protruding from the external lipid layer.

### 3.2. Characterization of the Aβ Peptide

The aggregation kinetics of the Aβ_1–40_ and Aβ_1–42_ peptides, at 50 μM and 10 μM, respectively, was monitored by Thioflavin T (ThT) emission in quiescent conditions and at 20 °C. The aggregation of both peptides resulted in a much longer time frame than the SAXS experiments also in the presence of LUVs. Appendix A shows the kinetics of Aβ_1–42_ at 37 °C and 15 μM, alone and in the presence of the liposomes B1 or B2 with GM1 (4%). The aggregation lag phase is of the order of tens of hours for the peptide alone, while it was reduced to a few hours in the presence of LUVs. The aggregation time was, however, much longer than the time of the SAXS experiments (3–4 min). Furthermore, the absence of Aβ aggregation during SAXS experiments was verified by superimposing sequential time course data. Though the aggregation kinetic studies performed in the presence of different LUVs did not lead to any clear correlations between the matrix components and different kinetic behaviors (data not shown), the evidence indicates that in the presence of any LUV the aggregation accelerates (Appendix A).

### 3.3. Aβ Interaction with Different Matrices

#### Comparison Aβ_1–40_-Aβ_1–42_

The SAXS spectra immediately after mixing LUV with Aβ peptides were recorded and then analyzed using a multi-Gaussian model, starting from the electronic profile obtained for the LUV alone. First, to confirm our previous results, we studied the effect of Aβ_1–40_ on the electronic density of the bilayer for LUV made of the BASE matrix without and with 2% mol/mol GM1 [13]. Figure 3 shows the SAXS data, and their analysis is reported for both peptides, Aβ_1–40_ and also Aβ_1–42_, added to the two different LUVs (a and b). In agreement with our previous results, the addition of Aβ_1–40_ to LUV containing GM1 perturbed the electron density of the bilayer (Figure 3b), while for Aβ_1–42_ an effect on the bilayer density profile was noticed, especially in the absence of ganglioside.

Figure 4 shows the study of the interaction of Aβ_1–40_ and Aβ_1–42_ with the matrices of the BASE composition and also containing 5% mol/mol SM (B2), as well as different amounts of GM1 in LUV. For GM1 lower than 3% mol/mol, no interaction was measured in terms of structural perturbation of the bilayer, for both Aβ peptides, while a minimal interaction was noticed for GM1 3% mol/mol. At the highest GM1 concentration, a perturbation in the internal leaflet was also noticed.

Table 2 reports the *s* values for nodes (minima) and peaks (maxima) and the intensity ratios relative to nodes and peaks (r_nodes_ and r_peaks_) for all SAXS spectra. Two nodes and two peaks can be determined from the spectra. The parameters, obtained by local data fitting with a polynomial curve, are reported to highlight the differences in the data sets. Table 2 also reports the bilayer thickness, *d*, calculated by the electronic density profile and defined as the distance between the two shells, inner and outer, by fixing a threshold for the electronic density *ρ* = 0.05.

## 4. Discussion

Analysis of the SAXS profiles for all the LUVs studied, leads to an asymmetric electron density profile for the lipid bilayers, with a lower internal contrast, due to an expected rougher arrangement of the polar head [24]. Differences in the SAXS spectra in the presence of GM1 reflect a dose-dependent increase in bilayer width, due to the presence of the sugar polar head protrusion of the ganglioside [29] (see Table 2).

Differently to Aβ_1–40_, the Aβ_1–42_ peptide appears capable of interacting with the B1 membrane made of POPC:POPS:Chol, without GM1 and SM. In accordance with this, it has been reported that Aβ_1–42_ monomers have an affinity for membranes with lipidic components in a disordered state, such as the B1 matrix in the conditions studied [30]; moreover, the catalytic role of cholesterol molecules for Aβ_1–42_ amyloid kinetics has also been highlighted [31], which could explain the interaction observed with SAXS. Furthermore, the fact that the concentration of Aβ_1–42_ is five-times lower than that of Aβ_1–40_ reinforces the importance of the result. In the other matrices studied here, cholesterol is always present but can be sequestered to interact with sphingolipid components, such as GM1 and SM, alone or together [32]. Only in the B1 matrix, cholesterol at the concentration used here should be homogeneously distributed in the phospholipid matrix, contributing to a disordered phase lipid environment.

### 4.1. Sphingomyelin

The ternary lipid composition (PC:Chol:SM) is a good model to mimic certain physico-chemical features of a natural membrane, where phase separations are induced by forming the so-called lipid rafts rich in cholesterol, SM, and gangliosides [33]. Therefore, SM in the matrix, along with cholesterol molecules, better mimics the raft-like membrane environment when GM1 is also present, compared to a SM-free matrix, such as the one used in our previous studies [7,13,20]. According to the fact that sphingomyelin has a higher transition temperature between an ordered gel and disordered liquid (T = 37 °C) than the monounsaturated phospholipids (in the liquid phase at T = 20 °C), the SAXS spectra are consistent with a more compact bilayer (see Figure 2 and Table 2).

SM is a peculiar lipid, capable of forming intra and intermolecular hydrogen bonding, involving the -3OH and -2NH donor functional groups. The presence of SM in a membrane changes the organization of the bilayer, as our data show, and also the ability of the membrane to interact with the approaching proteins [34]. In particular, the results suggest that SM could sequester cholesterol, leading to a more inert bilayer, in terms of the interaction with the peptide. This preferential dynamic sequestration is supposed to be the ground in the formation of rafts domains [11]. Therefore, the low affinity of Aβ_1–42_ for the B2 matrix (where SM is present), comparable to that of Aβ_1–40_ could be related to a structural change of the membrane, where cholesterol sequestration lowers the affinity of Aβ_1–42_ for the bilayer [30,33,35,36]. These results show that Aβ_1–42_ functions as a sensor that detects small membrane perturbations and changes in composition, while, from another perspective a model membrane matrix containing cholesterol homogeneously behaves as a sensor for Aβ_1–42_.

### 4.2. Mono-Sialo-Gangloside/Sphingomyelin/Cholesterol

The addition of the Aβ peptides in solution to the LUV sample caused deviations in the SAXS spectra that resulted, through the analysis, in changes in the electron density profile. Both GM1 and SM play a role in the interaction of the membrane with Aβ_1–40_ monomers. As well established in several studies, GM1 shows a specific interaction with the Aβ_1–40_ peptide [12,13,37]. Furthermore, it has been reported that SM favors the formation of oligomers close to the membrane; a process that is inhibited by the presence of GM1 at concentrations close to the ones studied here [9,10]. Aβ_1–42_ is known to exhibit greater instability than Aβ_1–40_, having two other residues from the APP transmembrane region. Therefore, Aβ_1–42_ tends to spontaneously nucleate at physiologic temperatures and quiescent conditions, by forming unstable oligomers/protofibrils on pathways in the amyloid process [38]. This could be the reason why a similar behavior with respect to Aβ_1–40_ was observed by SAXS at a five-times lower concentration regime.

Oligomerization steals monomers from the interaction with GM1, and on the other hand, the GM1–monomer interaction hinders oligomers formation [9]. The consequence of this is that the presence of both SM and GM1 weakens the interaction, as seen by SAXS. In fact, competition between the oligomer–SM interaction and monomer–GM1 interaction, and also an interaction between the SM and GM1 polar head, could explain the different bilayer perturbations obtained when both components are present. At different doses of GM1, slight changes are noted, especially in the presence of SM. In ternary matrices, therefore, the bilayer perturbation becomes difficult to detect, due to the interactive role of SM. In Figure 2c, the electron density obtained in the presence of GM1 suggests a lesser protrusion than in the matrix also containing SM, consistent with an interaction between SM and GM1. Below 20% mol/mol concentrations, GM1 has a known tendency to segregate, by forming clusters [39]. Clusters of varying sizes were measured, depending on the concentration of GM1 and the lipid composition of the matrix [40]. It has been reported, for matrices similar to those studied here, containing cholesterol and sphingomyelin, that GM1 nanodomains are in the disordered fluid phase [9,40,41]. This was observed in a range of GM1 2–8% mol/mol; therefore, close to the concentrations of GM1 under physiological conditions. GM1 was indicated as both a neuroprotective and an enhancer agent of amyloid aggregation [8,10]. A unifying view for the role played by GM1 in amyloid aggregation leads to the interpretation that, not only does the Aβ–membrane interaction depend on the concentration of the matrix components (SM, Chol and GM1) and phase of the lipid matrix, together with the properties of the GM1 nanodomains, but that the ratio Aβ/GM1 is also crucial. SM alone, but also clusters of high density of GM1, can induce Aβ oligomerization; an event necessary to have a conformational change to β-sheet structured oligomers, behaving as seeds for nucleation [8]. In this regard, under the conditions studied here, the Aβ/GM1 ratio is lower than the limit of 1:22 reported for the structural conversion to β structures, except for the samples of Aβ_1–40_ that interacted with the B1, B2 +1% GM1, and B2 +3% GM1 (1:0, 1:6, and 1:15, respectively); conditions that, according to Ikeda and collaborators [8], should lead to seeding. However, it must be considered that the absolute working concentrations in that work [8] were much higher than those reported here. In fact, the density of GM1 in the clusters also plays a fundamental role, together with the ratio Aβ/GM1. The fact that the nanodomains here are in the disordered phase and that GM1 is at low density, makes the seeding process unlikely, this being in agreement with the evidence that no aggregation was observed during the SAXS experiment [8,41].

The kinetics of aggregation for Aβ_1–42_, studied by monitoring ThT fluorescence in the presence of the studied matrices showed, however, that the aggregation process starts, but over a much longer time scale than the SAXS experiments, as reported in the Appendix A. Kinetic results in the presence of the different matrices did not reveal a clear correlation between lipid composition and aggregation time, generally showing an overall catalyzing effect due to the lipid surface, whatever the matrix composition. However, there was an indication that, especially in the presence of a matrix containing sphingolipids, the membrane system clearly makes the fibril formation process much less predictable, as shown by the standard deviation of the half growth time t_1/2_ (Appendix A).

## 5. Conclusions

The SAXS experiments and data analysis reported here successfully highlight differences in the bilayers of different lipid compositions. In particular, a dose-dependent effect of GM1 and the compaction of the matrix due to presence of SM, could be appreciated. The results show the fundamental role of cholesterol, capable of interacting with Aβ_1–42_ peptide, and of ganglioside GM1, interacting preferentially with Aβ_1–40_. The presence of sphingomyelin lowers both these interactions, by sequestering cholesterol and both Aβ and GM1; thus, being an efficient competitor in such interaction processes, due to its ability to create hydrogen bonds with surrounding molecules. Future biophysical studies using X-ray diffraction on ordered multilamellar systems, as well as neutron scattering, can help to shed further light onto the membrane–Aβ peptide interaction.

## Figures and Tables

**Figure 1 biology-11-00026-f001:**
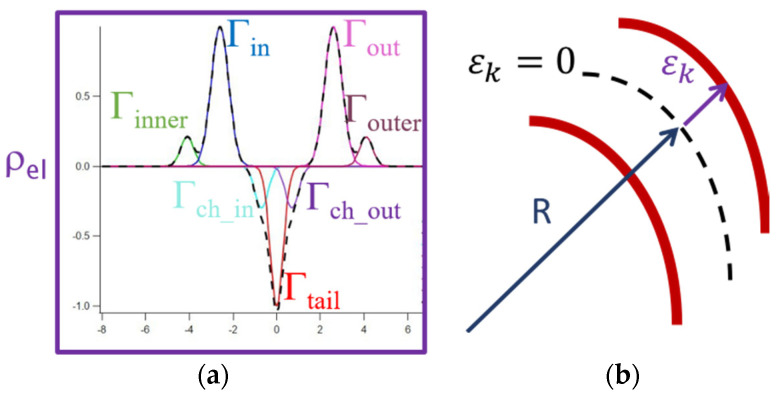
(**a**) Gaussians electron distribution (dashed line); different electron density contributions (continuous lines): inner decoration (green), internal polar head (blue), tail region (red), external polar head (magenta), outer decoration (brown) and cholesterol perturbations, i.e. ch_in (cyan) and ch_out (violet). (**b**) Vesicle radius, *R*, definition as the distance from the center of the vesicle to δ*_tail_*, i.e., the center of the bilayer.

**Figure 2 biology-11-00026-f002:**
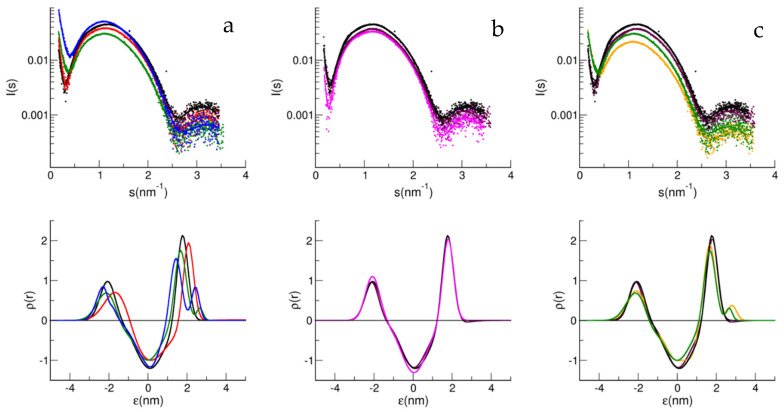
SAXS spectra (symbol) and fit curves (line) (upper panels): B2 with GM1 0% (black), 1% (red), 2% (green), and 4% (blue) (**a**); B1 (black), B2 (maroon) and B3 (magenta) (**b**); B1 (black), B2 (maroon) and B1 with GM1 3% (yellow), B2 with GM1 2% (green) (**c**). Electron density obtained from the multi Gaussian analysis (lower panels): B2 without (black line) and with GM1 1% (red line), 2% (green line), and 4% (blue line) (**a**); B1 (black), B2 (maroon line) and B3 (magenta line) (**b**); B1 (black line), B2 (maroon line) and B1 with GM1 3% (yellow line), B2 with GM1 2% (green line) (**c**).

**Figure 3 biology-11-00026-f003:**
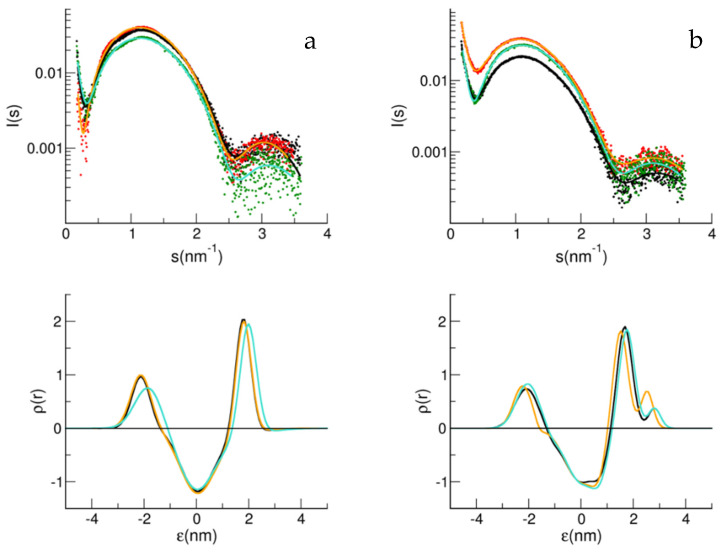
SAXS spectra and fit curves (upper panels) for B1 (**a**) and B1 with GM1 2% (**b**). LUV alone (black symbols for data and black line for fit curve), LUV with Aβ_1–40_ (red symbols for data and orange line for fit curve), and LUV with Aβ_1–42_ (green symbols for data and cyan line for fit curve). Electron density obtained from the multi Gaussian analysis (lower panels) for B1 (**a**) and B1 with GM1 2% (**b**). LUV alone (black line), LUV with Aβ_1–40_ (orange line) and LUV with Aβ_1–42_ (cyan line).

**Figure 4 biology-11-00026-f004:**
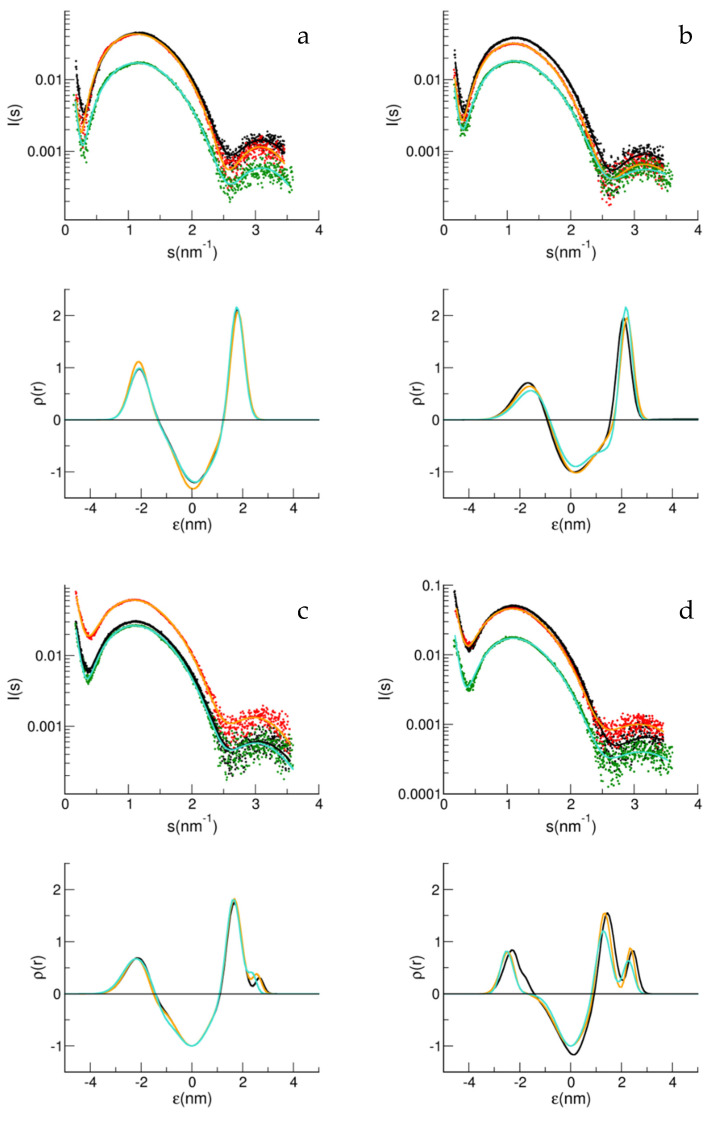
SAXS spectra and fit curves (upper panels) for B2 without (**a**) and with GM1 1% (**b**), GM1 3% (**c**) and GM1 4% (**d**). LUV alone (black symbols for data and black line for fit curve), LUV with Aβ_1–40_ (red symbols for data and orange line for fit curve), and LUV with Aβ_1–42_ (green symbols for data and cyan line for fit curve). Electron density obtained from the multi Gaussian analysis (lower panels) for B2 without (**a**) and with GM1 1% (**b**), GM1 3% (**c**) and GM1 4% (**d**). LUV alone (black line), LUV with Aβ_1–40_ (orange line) and LUV with Aβ_1–42_ (cyan line).

**Table 1 biology-11-00026-t001:** Composition of the lipid matrix of the LUV bilayers. The concentration of each component is expressed as % mol/mol. <M> is the average molecular mass. PC: phosphatidil-choline; PS: phosphatidil-serine; Chol: cholesterol; GM1: ganglioside; SM: sphingomyelin.

Name	PC	PS	Chol	GM1	SM	<M>
BASE (B1)	78	8	14	0	0	702
B1 + GM1 2%	75	7	16	2	0	718
B1 + SM 5% (B2)	73	7	15	0	5	703
B2 + GM1 1%	72	7	15	1	5	711
B2 + GM1 3%	70	7	15	3	5	727
B2 + GM1 4%	69	7	15	4	5	734
B1 + SM 10% (B3)	68	7	15	0	10	701

**Table 2 biology-11-00026-t002:** Position of the two nodes and peaks and intensity ratio relative to nodes (r_nodes_) and peaks (r_peaks_).

	Node_1_ (nm)	Peak_1_ (nm)	Node_2_ (nm)	Peak_2_ (nm)	r_nodes_	r_peaks_	*d* (nm)
BASE (B1)	0.321	1.187	2.617	3.148	1.18	0.47	5.50
+Aβ_1–40_	0.289	1.165	2.581	3.107	1.17	0.50	5.61
+Aβ_1–42_	0.335	1.186	2.617	3.091	0.94	0.39	5.79
B1 + GM1 2%	0.397	1.089	2.652	3.135	1.25	0.39	6.39
+Aβ_1–40_	0.410	1.064	2.615	3.084	1.47	0.43	6.15
+Aβ_1–42_	0.386	1.095	2.663	3.177	1.17	0.45	6.37
B1 + SM 5% (B2)	0.311	1.172	2.626	3.114	1.16	0.46	5.57
+Aβ_1–40_	0.279	1.149	2.610	3.083	1.15	0.54	5.59
+Aβ_1–42_	0.301	1.168	2.605	3.134	1.16	0.46	5.55
B2 + GM1 1%	0.332	1.127	2.641	3.106	0.81	0.36	5.81
+Aβ_1–40_	0.312	1.103	2.611	3.099	0.73	0.33	5.91
+Aβ_1–42_	0.322	1.118	2.615	3.154	0.72	0.26	5.77
B2 + GM1 3%	0.376	1.102	2.616	3.193	1.28	0.39	6.26
+Aβ_1–40_	0.395	1.077	2.587	3.131	1.31	0.37	6.21
+Aβ_1–42_	0.376	1.103	2.618	3.089	1.33	0.37	6.10
B2 + GM1 4%	0.405	1.097	2.637	3.101	1.6	0.54	6.06
+Aβ_1–40_	0.409	1.066	2.622	3.077	1.86	0.52	5.95
+Aβ_1–42_	0.400	1.089	2.619	3.067	1.95	0.67	5.86

The last column reports the distance d between the inner and outer shell, obtained by fixing a threshold for the electron density at = 0.05.

## Data Availability

Data are available under request to the authors.

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
