# Peer review of "Small Angle X-ray Scattering Sensing Membrane Composition: The Role of Sphingolipids in Membrane-Amyloid β-Peptide Interaction"

_biology, 2021, doi:10.3390/biology11010026_

Round 1

Reviewer 1 Report

The new reported innformation on the interactions of amyloid beta is valuable,

the study is sound and the presentation is quite good, though a bit technical,

but with the merit of precision.

The presented interplay between amyloid-beta interactions with cholesterol, ganglioside GM1 and sphingomyelin can be vauable in the mechanistic understanding of plaque formation in Alzheimer's and potential drug action.

Author Response

We thank the reviewer for its positive comments.

Reviewer 2 Report

In this manuscript, authors have used Small angle X ray scattering (SAXS) to describe the interaction between Aβ peptide and large unilamellar vesicles (LUV) of different lipid combinations. Most of the conclusions derived from the study were already preexisting in the literature, still authors tried to quantify them in structural terms. 

I would like to ask fo some statistical analysis describing how significant are the shifts in the SAXS profile in relevant figures.

Are there any other biophysical methods or electron microscopy studies that can be used for better resolution to strengthen the conclusions? Authors should comment on that, and even better if they can provide some electron microscopy data which in my opinion would make the paper stronger.

The novelty of the paper is hard to follow since there are no illustrative figures to describe the model that authors are trying to propose. I strongly suggest to add atleast one such figure to help readers.

Also, the paper should be revised thoroughly for English corrections. 

Reviewer 3 Report

The study has been evaluated the role of sphingolipids on membrane-amyloid beta-peptide interaction through small-angle X-ray scattering sensing membrane composition. However, a few minor issues need to be addressed:

Page 1; Line 21: Abstract needs to be constructed in a way that all the outcomes with the significant figures should be given either in the percentage or statistically significant values.

Page 11; Line 377: Conclusion should highlight and use of the provided method for future study.

Author Response

Answer to Reviewer 3

The study has been evaluated the role of sphingolipids on membrane-amyloid beta-peptide interaction through small-angle X-ray scattering sensing membrane composition. However, a few minor issues need to be addressed:

Page 1; Line 21: Abstract needs to be constructed in a way that all the outcomes with the significant figures should be given either in the percentage or statistically significant values.

In the paper we included a table (Table 2) with some parameters derived from the data spectra to give the extent of changes by effective percentage. Moreover, the table reports an estimate from the model based analysis of the membrane thickness, d, in order to understand how spectra changes pair to model parameters. We included the relevant ones in the abstract.

Page 11; Line 377: Conclusion should highlight and use of the provided method for future study.

A perspective for our future work, related to our study, has been added in the conclusions.

Round 2

Reviewer 2 Report

Thanks to the authors for addressing my concerns regarding the manuscript. I congratulate the authors for the effort and recommend for quick publication of the manuscript.